# 3D-Printing of Hierarchically Designed and Osteoconductive Bone Tissue Engineering Scaffolds

**DOI:** 10.3390/ma13081836

**Published:** 2020-04-13

**Authors:** Nicolas Söhling, Jonas Neijhoft, Vinzenz Nienhaus, Valentin Acker, Jana Harbig, Fabian Menz, Joachim Ochs, René D. Verboket, Ulrike Ritz, Andreas Blaeser, Edgar Dörsam, Johannes Frank, Ingo Marzi, Dirk Henrich

**Affiliations:** 1Department of Trauma, Hand and Reconstructive Surgery, University Hospital, Goethe University Frankfurt am Main, 60590 Frankfurt, Germany; Jonas.Neijhoft@kgu.de (J.N.); Rene.Verboket@kgu.de (R.D.V.); Johannes.Frank@kgu.de (J.F.); marzi@trauma.uni-frankfurt.de (I.M.); d.henrich@trauma.uni-frankfurt.de (D.H.); 2Department of Mechanical Engineering, Institute of Printing Science and Technology, Technical University of Darmstadt, 64289 Darmstadt, Germany; nienhaus@idd.tu-darmstadt.de (V.N.); valentin.acker@gmail.com (V.A.); jana.harbig@googlemail.com (J.H.); fjmenz@gmail.com (F.M.); jochs-taunus@gmx.de (J.O.); doersam@idd.tu-darmstadt.de (E.D.); 3Department of Orthopedics and Traumatology, Johannes Gutenberg-University Mainz, 55131 Mainz, Germany; Ulrike.Ritz@unimedizin-mainz.de; 4Institute for BioMedical Printing Technology, Technical University of Darmstadt, 64289 Darmstadt, Germany; blaeser@idd.tu-darmstadt.de

**Keywords:** Bone Tissue Engineering, smart scaffold, scaffold design, osteoconductive

## Abstract

In Bone Tissue Engineering (BTE), autologous bone-regenerative cells are combined with a scaffold for large bone defect treatment (LBDT). Microporous, polylactic acid (PLA) scaffolds showed good healing results in small animals. However, transfer to large animal models is not easily achieved simply by upscaling the design. Increasing diffusion distances have a negative impact on cell survival and nutrition supply, leading to cell death and ultimately implant failure. Here, a novel scaffold architecture was designed to meet all requirements for an advanced bone substitute. Biofunctional, porous subunits in a load-bearing, compression-resistant frame structure characterize this approach. An open, macro- and microporous internal architecture (100 µm–2 mm pores) optimizes conditions for oxygen and nutrient supply to the implant’s inner areas by diffusion. A prototype was 3D-printed applying Fused Filament Fabrication using PLA. After incubation with Saos-2 (Sarcoma osteogenic) cells for 14 days, cell morphology, cell distribution, cell survival (fluorescence microscopy and LDH-based cytotoxicity assay), metabolic activity (MTT test), and osteogenic gene expression were determined. The adherent cells showed colonization properties, proliferation potential, and osteogenic differentiation. The innovative design, with its porous structure, is a promising matrix for cell settlement and proliferation. The modular design allows easy upscaling and offers a solution for LBDT.

## 1. Introduction

Treatment of critical size bone defects remains a major challenge in modern traumatology/orthopedics [1]. Disadvantages and complications of established procedures make treatment alternatives urgently necessary [2,3]. Bone Tissue Engineering is considered to be a key technology to overcome current limitations [4,5]. The ideal bone substitute (scaffold), once introduced into the osseous defect, recruits osteogenic and angiogenic stem cells (osteoconductive), navigates cell differentiation and finally stimulates bone and vascular formation (osteoinductive and angiogenic properties). Moreover, the resorbable scaffold should mechanically stabilize the defect zone for this period, until it is completely replaced by newly formed, autogenous bone.

To meet the desired properties, a scaffold should be mechanically stable, absorbable, osteogenic (osteoconductive and osteoinductive), angiogenic, and cytocompatible (non-immunogenic, have no toxic degradation products, and stimulate no foreign body reaction) [6].

While promising scaffold solutions for critical size defect treatment in small animals and in vitro exist, successful applications in large animal models have only occasionally been documented and significant innovations for commercially available bone substitutes do not exist [6,7,8,9,10,11].

According to Leijten et al. (2015), one reason is that many researchers consider only partial aspects in their conceptions of bone substitutes. Instead of rigidly imitating bone characteristics, the entire process of dynamic bone healing has to be taken into account [12]. Especially the initial inflammatory and granulation phase including hematoma formation and the immigration of cells and vessels is often neglected. Immigrating cells must be supplied with nutrients and oxygen by diffusion until sufficient vascularization is restored [13,14,15,16]. For central located cells—far away from peripheral zones with intact diffusion-mediated nutrition and oxygen supply—risk of poor supply is particularly high [15,17,18].

Distances of a few hundred microns maximum with large gradients are necessary for rapid mass transfer via diffusion [19]. This requires an open and porous scaffold design with cavity diameters of a few millimeters [14,15,20]. The invading hematoma as a matrix for the invading macrophages and stem cells requires large pores (>2 mm) to penetrate the depths of the scaffold [6]. Contact of blood with thrombogenic foreign material—in this case, the scaffold—could immediately lead to coagulation [21]. If this occurs, the flow properties of blood change fundamentally. Small pores quickly become a barrier. In current developments, the existing pore diameters seldom exceed 1 mm [11,22,23,24,25,26,27,28,29,30]. Open structures of several millimeters are mostly neglected [31]. This is explained by the decreasing stability which occurs with increasing porosity [32,33,34,35]. The hematoma no longer reaches central scaffold areas due to narrow and long channels, which usually arise when adjusting common scaffolds to long-range defect distances (upscaling). One possible consequence is marginal bone formation without bone formation in areas close to scaffolds center [10]. However, the requirement for rapid penetration of blood is counteracted by the need for finely-textured surfaces that favor cell attachment and the formation of a regenerative microenvironment. The dimensions of fine structures have very differentiated biological influence. According to their biological effect, microfilamentary, microporous, and macroporous geometries can be distinguished. Microporous/filamentary structures have a major influence on cell adhesion. The high surface/volume ratio and microporosity of these structures favor cell/material interaction via adhesion proteins/polysaccharides. Thus, higher cell densities are possible [22,23,24,25,26,36,37]. Macroporosity refers to pores of at least 100 μm in diameter. For cell–cell interactions, a communicating pore system is essential. A porosity of more than 90% is optimal [38]. The pores provide a stimulating effect on osteoconductivity and osteoinductivity. Pores 300–500 μm in diameter stimulate the formation of new blood vessels with a sufficiently large lumen. Only pores below 100 μm increase the probability of hypotrophic regeneration with cartilage tissue [39]. However, the problem is that high surface/volume ratios, small diameters, and filamentary fine structures can quickly contribute to a closure of the inner lumen of scaffolds.

The aim of this “proof of concept” study was to develop a novel scaffold system based on PLA, which especially supports the initial phase of bone regeneration through an innovative three-dimensional architecture and, moreover, can be easily adapted to larger defect dimensions. The focus is on a hierarchical approach over five levels. Small, biologically functional units are embedded in a load-bearing, compression-resistant frame structure, thus ensuring mechanical stability on the one hand, and diffusion of oxygen and nutrients into the inner areas of the implant via open architecture and pores on the other. The most promising designs were printed and characterized mechanically and in vitro.

## 2. Materials and Methods

A scaffold prototype was developed and realized using Fused Filament Fabrication (FFF), a common additive manufacturing technology. It utilizes a thermoplastic wire (filament) which is melted and extruded layer-wise to form a part. After determination of the mechanical properties, cell adhesion, viability, osteogenic differentiation, and cell morphology of Saos-2 cells seeded on the scaffolds were evaluated in vitro.

### 2.1. Scaffold Design

The scaffold was designed using Fusion 360 (Autodesk, San Rafael, CA, USA), a 3D design software which can be used for computer-aided design (CAD) and rapid prototyping. It was designed with constraints and parameters, allowing for later customization using a mix of boundary representation (BREP) and constructive solid geometry (CSG).

### 2.2. Scaffold Fabrication

The designs were printed on a modified standard FFF-printer (ANET A6, Shenzhen Anet Technology Co., Shenzhen, China). The stl-files were processed using the slicing software Cura Version 3.1.0 (Ultimaker, Geldermalsen, Netherlands) into a coordinate-based file format called G-code. These G-codes were loaded onto a Raspberry Pi 3b+ (Raspberry Pi Foundation Cambridge, United Kingdom) running with the opensource system Octoprint to control the 3D printer. The print bed covered with a fiberglass FR4-plate was heated at 60 °C. A filament with a diameter of 2.85 mm (Design 1)/1.75 mm (Design 2) made of PLA (for biological experiments undyed PLA was used, while dyed PLA was used for design illustration) (3dk.berlin, Berlin, Germany) was extruded through a 100 µm (Design 1)/200 µm (Design 2) wide nozzle heated at 200 °C, if not described differently. While the basic print speed was set to 40 mm/s, depending on the scaffold´s contact area with the bed, a raft or skirt was used for better adhesion. Then, the scaffold was produced with a layer height of 100 µm and retraction distances of 5 mm. To disinfect the scaffold, it was immersed in 70% vol. alcohol for 10 min and subsequently dried for 10 h in a 12 well plate.

### 2.3. Examination of Blood Penetration and Hematoma Formation

Thirty milliters of peripheral blood without anti-coagulatory agents were obtained from one of the authors (N.S.) and immediately subjected to two wells of a six well-plate (Nunc, Wiesbaden, Germany). Blood clotting was allowed over a period of 3 min and scaffolds (two connected elements of Design 1 and a single scaffold of Design 2) were carefully placed laterally into the blood for a time period of 30 s. Then, scaffolds were carefully removed using forceps and placed in individual empty wells of a six-well plate, thereby keeping their initial orientation. After 3 h incubation at room temperature to complete coagulation, the frontal element of Design 2 was carefully removed using a scalpel in order to provide inside view.

### 2.4. Saos-2 Cell Cultivation

For biological evaluation, Saos-2 cells were purchased from DSMZ (Braunschweig, Germany, ACC 243). Cells were cultivated in a 75 cm^2^ culture flask (Sarstedt, Nümbrecht, Germany) in RPMI medium 1640 (Life technologies, Carlsbad, USA) supplemented with 10% Fetal Bovine Serum (Life technologies, Carlsbad, USA), 2% Hepes, 100 IU/mL penicillin, and 0.1 mg/mL streptomycin (both SIGMA, Deisenhofen, Germany) at 37 °C and 5% CO_2_. The medium was changed twice a week. After confluency reached 75%, cells were harvested. Non-adherent cells and debris were removed by washing with 3 mL of Dulbecco´s Phosphate Buffered Saline (−/− PBS without Mg^2+^ and Ca^2+^, Invitrogen, Bleiswijk, Netherlands). For experiments, cells were detached using accutase treatment (10 min, SIGMA-Aldrich, St. Louis, MO, USA) and this was followed by washing in RPMI medium 1640.

### 2.5. Scaffold Functionalization, Cell Adhesion, and Cell Seeding

A cell seeding protocol was developed, and efficiency was controlled by fluorescence microscopy and scanning electron microscopy (SEM). To enable cellular adhesion, the PLA scaffolds were functionalized applying a thin coating of 15.9 µg/mL BD *CellTak* Adhesive (BD Biosciences, Franklin Lakes, USA) in −/− PBS. For this, scaffolds were incubated for 30 min in the *CellTak* solution and subsequently air dried for 30 min. Before seeding, Saos-2 cells were stained with CFSE following the instructions of the manufacturer (Molecular probes, Thermo-Fisher, Schwerte, Germany). Subsequently, 10^6^ cells were seeded by slowly dripping cell suspension three times over the scaffolds, followed by 10 min of incubation. The process of dripping and incubating was repeated two times. Cells were washed twice with −/− PBS, fixed in 1% formalin in −/− PBS for 10 min and washed three times with −/− PBS. Counterstaining of the nuclei was accomplished by adding 1% of DAPI (10 µg/mL, 10 min, 4′,6-diamidino-2-phenylindole, Sigma-Aldrich) mixed with −/− PBS. Five-minute incubation was followed by washing four times with −/− PBS. Adhering cells were then detected by means of fluorescence microscopy (AxioObserver Z1, Carl Zeiss, Göttingen, Germany). The number of adherent Saos-2 cells was counted on randomly chosen high power fields at 100-fold magnification. Furthermore, the bare surface of the scaffolds and those seeded with cells were analyzed using scanning electron microscopy (SEM). Samples were fixed with glutardialdehyde (2% in −/− PBS) for 10 min and were immersed in a four-step ethanol gradient (50%, 75%, 96%, and 100%) for 5 min each. After a short passage in 1,1,1,3,3,3-hexamethyldisilazane (Merck-Schuchardt, Hohenbrunn, Germany) and draining overnight, gold was deposited on the samples by sputtering (5 × 60 s, Agar Sputter Coater; Agar Scientific Ltd., Stansted, United Kingdom). Analysis was performed using a scanning electron microscope (Hitachi, Düsseldorf, Germany) and the Digital Image Processing System (DIPS) 2.6 (Point Electronic, Halle, Germany).

### 2.6. Cell Viability

Saos-2 viability on the scaffolds was evaluated by use of the MTT Cell Proliferation Kit 1 (Roche Diagnostics, Mannheim, Germany), in which metabolic active and living cells reduce the yellow tetrazolium salt 3-[4,5-Dimethylthiazol-2-yl]-2,5-diphenyltetrazolium bromide (MTT) to purple formazan. Saos-2 cells were seeded on the scaffolds and fed with fresh medium every 24 h. Prior to cell seeding, the scaffolds were functionalized by *CellTak* adhesive as described previously. After 1, 7, 14, and 21 days, MTT assays (n = 3 per scaffold) were performed. Scaffolds were transferred to an empty well and then 3600 µL Medium and 400 µL MTT labeling reagent (10% MTT in −/− PBS) were added. Following 4 h incubation, a solubilization solution was added, and it was incubated overnight. Subsequently absorbance at 570 nm was measured using a plate photometer (Ceres UV900c, Bio-Tek Instruments, Windoski, VT, USA) and normalized to a calibration curve with 5, 10, and 15 × 10^6^ Saos-2 cell 2D monolayer, which was cultured in a sterile well of a 12 well plate in the absence of the *CellTak* adhesive.

### 2.7. Osteogenic Differentiation

Osteogenic differentiation was evaluated by Alizarin staining, which detects calcium depositions and expression of osteogenic marker genes by means of RT-PCR. For osteogenic stimulation, Saos-2 cells were seeded to the scaffolds and incubated over 7 days in standard culture medium followed by 14 days incubation in standard medium supplemented with 2.5% fetal bovine serum, 5% Hepes, 100 IU/penicillin, 0.1 mg/mL streptomycine, 0.2 mg/mL ascorbic acid, 0.1 mol/mL β-glycerophosphate, and 0.1 µmol/mL Dexamethasone (STEMCELL technologies, Cologne, Germany, sold as kit).

#### 2.7.1. Alizarin Red Staining

To detect calcium deposited by osteogenic differentiated cells adherent to the scaffold surface, alizarin red staining was applied. For this assay, the single cylinder subunit (Level 3) was used. Alizarin red staining solution (Sigma-Aldrich, Deisenhofen, Germany, TMS-008) was adjusted to pH 6.36–6.4 by adding 0.28% ammonia solution. Medium was removed and cells were fixed with 1% formalin in −/− PBS. Then, alizarin red staining solution was added and incubated for 10 min at room temperature. After removal and washing of the scaffolds, three times with distilled water, calcium deposits were evaluated using bright field microscopy.

#### 2.7.2. Gene Expression Analysis

RNA was isolated using *RNeasy* (Qiagen) following the manufacturer´s instructions. RNA concentration was measured and checked using a NanoVue 4282 V2.0.4 Spectrophotometer (GE Healthcare, Chicago, USA). Contaminating DNA was removed with a RNase-Free DNase Kit (Qiagen, Hilden, Germany) following the manufacturer´s instructions followed by first strand cDNA synthesis using an AffinityScript PCR cDNA Synthesis Kit (Stratagene, La Jolla, USA). Subsequent to cDNA synthesis, real-time PCR was performed using specific primers for collagen-1 alpha (Col1α, Qiagen, Hilden, Germany, NM_000088, catalog number, PPH01299F), alkaline phosphatase (ALP, Qiagen, Hilden, Germany, NM_000478, catalog number PPH00643F), bone gamma-carboxyglutamate protein (BGLAP, Qiagen, Hilden, Germany, NM 199173, catalog number, PPH01898A), and glyceraldehyde 3-phosphate dehydrogenase (GAPDH, Qiagen, Hilden, Germany, NM_002046.3, catalog number, PPH00150E) as reference gene. Pure medium was used as negative control. The thermal profile was 10 min at 95 °C, followed by 40 cycles of 15 s at 95 °C and 1 min at 60 °C. The obtained threshold cycle (Ct) values were evaluated using the Livak method (2^−(ΔΔCt^) [40].

### 2.8. Evaluation of Compressive Strength

Axial compressive strength tests were performed according to modified DIN EN ISO 604 protocol using a Zwick Z050 (ZwickRoell, Ulm, Germany). The test speed was 1 mm/s. The failure criterion was the abrupt decrease of the compressive force due to the collapse of the scaffold.

### 2.9. Statistics

All results are presented as mean values and standard error of mean (StdEM).

## 3. Results

### 3.1. Design and Fabrication of a Hierarchical Bone TE Scaffold

This work comprises the design, fabrication, and biological characterization of an innovative, hierarchical, and modular bone tissue engineering concept. The framework has a micro-, meso-, and macrostructure that is not only potentially adapted to support bone healing, but also focuses applicability and surgical use under clinical conditions. The design concept comprises five levels and is organized hierarchically (Table 1). Addressing optimized cellular ingrowth and nutrient supply, the first four levels differ in pore size and microstructural morphology. Level 5 enables modular connectivity of subunits, which can be assembled individually by the surgeon to match the final scaffold dimensions to the size of the bone defect. Starting from micro- to macroscopic architecture, the individual levels of design are described in Table 1.

### 3.2. Scaffold Architecture, Porosity, and Mechanical Properties

Level 1 is characterized by single strand dimensions and its surface microporosity (0.1–10 µm), which supports the cell-material interactions of immigrated cells (Figure 1i,j).

Level 2 (5 mm diameter, 2 mm inner hole, and 0.15 mm height) includes filamentary networks with pore sizes of 100–150 µm (Figure 1a,b). The cylindrical ring is filled with strands in a zigzag grid with a 100–150 µm spacing. The grid is created as an automatic infill by the slicer software Cura. The workflow was previously described by Nienhaus et al. [41]. Inside the inner hole, an additional network of fine filaments is located (Figure 1a,b). It was not achieved by controlled extrusion with standard printing parameters that, for instance, were used for the porous outer walls. Instead, a novel printing process was applied enabling fabrication of ultrafine filaments and filament networks that resemble electron-spun matrices (Figure 1a). Briefly, by varying the process parameters material flow and printing speed (Table 2), it was finally possible to achieve stringing structures with a controllable path. Even though the accumulation of filaments cannot be reproduced visually, the number and diameter of the filament is reproducible (Figure 1a). A first approach to this technique was described by Siebert et al. [42] These structures offer the desired high surface/volume ratio for cell adhesion in close proximity to micro- and macroscopic pores in the wall of the scaffold (Figure 1c,d).

On Level 3, the filamentary networks were stacked and additional larger pores were added. The structure has a diameter of 5 mm and measures 5 mm in height. It is created as a solid volume. The slicer automatically fills the structure with incrementally rotating grids of the Level 2. Axially, each cylinder starts and ends with a base ring of two solid layers (Figure 1d). The walls in between are printed as a zigzag grid with mesh sizes of 100–150 µm (Figure 1c,d). For better illustration of the wall’s porous fine structure, the solid base ring has been left out in Figure 1b. The cylinder is punctured by holes of 700–1500 µm diameter (Figure 1c,d). For mechanical stability, an additional solid support structure was superimposed (Figure 1d). The results of a compression test of single hollow columns with helical structure, octagonal truss, and a combination of both is given in Figure 1k. The truss has a good mechanical stability and occupies less space than the combination of helix and columns. The compressive strength can be largely influenced by the strut diameter. In this case, a strut diameter of 0.6 mm was chosen as a compromise between compression strength and reduction of solid volume.

On the fourth scaffold modularity level, two different scaffold designs were developed and tested. First, an annular structure with a continuous central canal measuring 8 mm in diameter and eight rotationally symmetric hollow cylinders of Level 3 were fabricated. (Figure 1e). Besides additional porosity, the eight hollow cylinders served as anchors to fix the scaffold in a femur defect in the future. To enable manual mounting of multiple, individual scaffold subunits, a connection system was developed using external brackets (Figure 1d,e). The end caps were intended to anchor in the medullary canal.

The second design comprised a highly porous structure with a central vertical and horizontal channel as well as four rotationally symmetric hollow cylinders (Figure 1a–c). The structure consisted of columns functioning as subunits, which were combined by means of a small pedestal with a ring on the bottom and the top. The diameter of the columns was 4.5 mm with a thickness of 0.4 mm. Pores with sizes of 1 and 1.5 mm were added, while additional microporosity was generated by the intercolumn space and printing process itself during cooldown of PLA (Figure 1i,j). In the middle, there was a reservoir-like structure and a hexagonal connecting adapter (Figure 1g). The scaffold had a calculated surface area of 4438.89 mm^2^ and volume of 796.93 mm^3^. It weighted 0.760 (±0.028) g.

### 3.3. Examination of Blood Penetration and Hematoma Formation

Scaffold prototypes are quickly saturated with blood. To analyze the penetration of blood into the internal scaffold structures, the scaffold prototypes were immersed for 30 s in peripheral blood that was not provided with anticoagulants to simulate the situation in a freshly treated bone defect. The investigation showed that, in principle, inner and outer surfaces were completely covered by blood. After 3 h incubation, hematoma formation in both scaffold in all areas could be detected (Figure 2).

### 3.4. Biological Characterization Hierarchically Designed Bone TE Scaffolds

According to the defined design criteria, bone TE scaffolds were fabricated and biologically characterized. For cell experiments, exclusively undyed PLA scaffolds were used. Cell adhesion experiments were performed with Saos-2 cells. Smooth PLA surfaces had to be preconditioned with *CellTak* to enhance cell adhesion (Figure 3c–e). After preconditioning especially in the intercolumn spaces, stringing nets and cooling cracks in the PLA scaffold areas of high cell density were detectable (Figure 3a,b). The scaffold was penetrated deeply and showed cell adhesion even at the central filamentary structures (Figure 3f,g). At the same time, an increase in cell activity from Day 1 to Day 21 could be observed, as shown in the MTT-assay (Figure 3h). In addition, Alizarin red staining indicated that cells cultured under osteogenic differentiation media left calcium depositions on the scaffolds (Figure 4a,b). Gene expression analysis underlined these findings. Gene expression on Day 21 showed a 1.37-fold increase of collagen-1-alpha compared to control and a 12.78-fold increase of alkalic phosphatase compared to control. A BGLAP (0.48-fold increase compared to control) elevation could not be detected.

## 4. Discussion

The aim of this work was the development and in vitro testing of novel scaffold designs for long-bone-defect treatment, which potentially activate all phases of secondary bone healing, especially the initial phase.

### 4.1. Process Development

The complexity of the designed scaffolds with microscopic and macroscopic internal structures makes a high-precision and reproducible manufacturing process necessary. These requirements can currently only be met by *additive manufacturing*. Workpieces for this *“proof of concept”* study were made of PLA.

Production of filamentary nets with mesh sizes of less than 100–150 μm using conventional 3D-printing methods is challenging. The extruded filament has a diameter of 0.1 mm/0.2 mm. Thinner strands are preferable for such structures. Preliminary tests showed that controlled extrusion is not suitable for these structures. In contrast, the unwanted stringing structures in conventional applications are promising. Depending on printing parameters, strands of a few micrometers in diameter can be produced. Currently available software packages for slicing CAD models are designed to explicitly avoid stringing. Thus, it was necessary to find a way in which stringing can be controlled and reproducibly enforced. The presented printing parameters (Table 2) provide the possibility of printing filamentary structures within a range from less than 1 to 100 µm in reproducible number and density as required.

The choice for a cylindrical scaffold structure was based on two considerations. On the one hand, the physiological bone cross-sectional area was taken into account. On the other hand, technical aspects were included. Basically, angular geometries lead to changes of direction in the nozzle path. These changes of direction caused the printer to vibrate, which in turn reduced component quality. In addition, corners could not be reproduced with acceptable tolerance due to abrupt deceleration of one or more axes associated with them.

A truss structure with a strut thickness of 0.6 mm was developed as the load-bearing element. This provides a compromise between strength and occupied volume. Axially, the cylinder had a 2 mm vent to ensure enough space for the penetrating hematoma. However, with this wall construction, the necessary pores in the ranges of >300 μm and <1 mm could not be realized. The solution here was to fill the areas between the struts with porous, filamentary wall structures (Figure 1c,d). The individual hollow column subunit thus offers all biologically relevant features and could be tested individually in the established small animal model, in the future [43,44,45,46]. Transplantation of a similarly dimensioned, 3D-printed, porous hollow column structure into a plate-stabilized 5 mm defect in the rat femur by two authors of this study showed promising healing results [47]. Transfer to a large animal model is conceivable, since current data indicate that scaffolds based on 3D printed fused filament fabrication can be effective for the treatment of large bone defects in large animals [9,48,49,50]. However, there is almost no structural comparability of the used scaffolds in large animals compared to the structures presented here.

Thus, one focus of this “proof of concept” study was the development of a design that potentially allows easy upscaling to larger bone defect cross-sections while maintaining scaffold integrity.

To maintain the structural integrity of the single column, an adaptation to larger dimensions should be achieved by combining multiple single columns on a common base (Figure 1d,e).

For the modular connection of the final scaffold, different connection systems were designed. Approximation to the final solution was an iterative process. For different designs, individualized printing protocols were necessary. Finally, a system with external or internal click brackets proved to be optimal regarding to rotational stability, flexural strength, torsional stability, printability and effect on the bioactive structures (Figure 1d,e). For connectivity, the brackets were combined with the lightweight panels and printed as solid structures. To avoid sharp edges, the transitions between the plates and the plugs were rounded off.

### 4.2. Scaffold Design, Mechanics, and Biological Aspects

The here tested scaffold served as a *“proof of concept”* to reconcile printing technology requirements with the biological-physical requirements.

While channels in the millimeter range essentially support hematoma penetration and formation, fine structure dimensions take very differentiated biological influence. According to their biological effect, microfilamentary, microporous, and macroporous geometries can be distinguished. Microporous/microfilamentary structures have major influence on cell adhesion. The high surface/volume ratio and microporosity of these structures favor cell–material interaction via adhesion proteins/polysaccharides. Higher cell densities are possible [22,23,24,25,26,36]. Macroporosity refers to pores of at least 100 μm in diameter. For cell–cell interactions, communicating pore systems are essential. A porosity of more than 90% is optimal [38]. The pores provide a stimulating effect on osteoconductivity and osteoinductivity. Pores of 300–500 μm in diameter stimulate the formation of new blood vessels with a sufficiently large lumen. Only pores below 100 μm increase the probability of hypotrophic regeneration with formation of cartilage tissue [39].

The fabricated multi-porous structure showed the required open architecture and geometrical features. Fine filament structures were introduced into the single hollow columns (Figure 1a). The network of single fibers in the range of several micrometers in diameter span a distance of about 2–3 mm and thus increase the reactive surface for cell attachment. These fine structures were in the immediate vicinity of macroscopic pores in the millimeter range (Figure 1c,d), since column walls are punctured by holes of 700–1500 µm diameter (Figure 1c,d). Furthermore, the hollow column wall is covered with a zigzag mesh with pore sizes from 100 to 150 µm. In combination with axial channels in the range of 2–4.5 mm, the single hollow column provides a high porosity with perpendicular and axial channels, although not all channels/pores are pores in the literal sense of the word. Maximum diffusion distances of 2 mm were not exceeded. Due to the relatively wide lumen, it is highly likely that tissue ingrowth is possible. The predominant type of tissue (bones, fibers, and/or vessels) must be determined by appropriate animal experiments. Spaced assembly of several hollow columns in the final design facilitate bleeding into the whole scaffold areas, and this subsequently allows hematoma formation within the complete scaffold (see additional experiment in Figure 2). Overall interfilamental stress cracks and sporadic stringing structures occurred. These provide an increasing surface/volume ratio and thus support cell adhesion (Figure 1i,j).

With pore interconnection of more than 90%, all structures functioned beyond the requirements (>90%). The introduction of fine structures as binding sites for cells is one of the key features in smart scaffold development.

In mechanical tests, a single hollow column subunit yielded approximately 150 N axial load, exhibiting a mechanical resistance of up to 30% of the axial dynamic loading of native femur [51,52]. Although such compressive strength and even more are also evident in other composite materials, these scaffolds, in most cases, offer significantly lower porosity [53]. Mineral scaffolds achieve such values only with significantly reduced porosity [54]. Only an axial compression test was carried out, since the relevant force transmission to the scaffold is in the axial direction [51]. Torsional and bending forces should be neutralized by the additionally introduced plate osteosynthesis. Basically, scaffolds should act as temporary placeholders until complete bone healing has happened. A complete bone replacement with all its physical and biological properties was never sought. Ideally, the scaffold degrades within the same time bone formation takes place. However, the current data allow no statement about scaffolds degradation. To analyze degradation, longer-lasting animal studies are necessary. The transplantation of a similarly dimensioned, 3D-printed, porous tube structure into a plate-stabilized 5 mm defect in the rat femur by two authors of this study showed signs of degradation of the implanted PLA scaffold after eight weeks of healing time [47]. It is concluded that, for a sufficient analysis of the degradation, a standing time of at least six months is necessary.

Scaffold colonization attempts were made with Saos-2 cells. These are similar in their colonization behavior to MSC, but much easier to handle. Despite the proven microstructure (Figure 1i,j) (stress cracks, interfilamentary angular spaces, and filament thickness of about 100 μm), a coating with adhesion protein was necessary on the scaffolds for colonization experiments. Preliminary experiments showed that cells found insufficient support on the native surfaces of the extruded PLA. In future steps, the PLA will be replaced by a PLA/BG composite. This material offers good cell adhesion properties, especially for osteogenic and angiogenic cells like MSC and EPC [55,56].

After scaffold coating, dense cell colonization over the entire scaffold surface was evident in the SEM images (Figure 3a,b). Due to high porosity of both structures, cells were able to penetrate deeply into the scaffolds (Figure 3d–g). Additionally, stringing structures were well reached and populated (Figure 3g,f); however, it must be taken into account that the cell suspension used (cells and medium) was significantly less viscous than blood. No coagulation or similar behavior was detectable. Thus, a complete penetration of the scaffold with cell suspension was guaranteed within the observation period. Whether blood is also able to completely penetrate the scaffold and form a hematoma was examined by an additional wetting experiment. Our experiments demonstrated that the open structure of the scaffold allows the rapid and complete influx of blood. The fracture hematoma, induced by the surgical treatment, therefore might also form within the scaffold. This might be important for a future use of this scaffold class for the treatment of large bone defects. The complete filling of the scaffold with the hematoma provides growth factors released by platelets and leukocytes entrapped within the fibrin network. Relevant factors are insulin-like growth factor-1 (IGF-1), which supports osteogenesis, and vascular endothelial growth factor (VEGF), which provides chemotactic and proangiogenic activity [57].

As expected, the high metabolic cell activity for all structures indicates generally good cytocompatibility of the PLA (Figure 3h) [58]. pH values in the cell culture medium were not determined. However, it is assumed that culture medium buffered pH to the physiological values.

The ideal scaffold should initiate osteogenic stimulation through the scaffold itself. Thus far, no osteoinductive effect has been demonstrated for pure PLA [59]. For this reason, it was necessary to add osteogenic substances into the culture medium to induce osteogenic differentiation of the Saos-2 cells. The alizarin staining revealed pronounced calcium precipitation by Saos-2 cells in osteogenic medium for all structures compared to cells in standard medium (Figure 4a,b). Additionally, it could be shown that applied cells were not impaired in their osteogenic differentiability after the addition of the osteogenic induction medium (Figure 4c).

## 5. Perspectives and Conclusions

The main hurdle for bone tissue engineering is the lack of scaffolds which sufficiently combine all the desired requirements. Since the current concepts did not lead to a breakthrough, new strategies are needed. Mimicking natural bone structure is not sufficient. Important aspects of bone healing, such as the initial phase when hematoma formation and hypoxic local milieu take place, must be considered and addressed. Promising concepts should ensure the supply of immigrating or already adherent cells throughout the scaffold and at all stages. Probably only then would concepts for stimulating cell differentiation and proliferation lead to success. Greater attention must therefore be paid to the geometric design of scaffolds allowing these biological steps. Previous publications have shown that individual structural elements can influence osteogenesis, angiogenesis, and resorption [18,23,26]. Now, it is important to combine these structural elements. However, it is only since the introduction of new additive manufacturing processes such as 3D printing that the implementation of complex, multi-layered designs has become possible.

The structural concepts presented here represent an innovative approach. The previous findings on the stimulation of angiogenesis and osteogenesis were incorporated into the design concepts in this paper. As Petersen et al. reported, the orientation of scaffold pores has a fundamental influence on cell invasion, angiogenesis, and finally direction of bone healing (intramembranous vs. endochondral bone healing) [60]. The design presented here offers the required axial pore orientation for cell invasion, as well as perpendicular pores. These could facilitate angiogenesis.

The mechanical and cell-biological results are promising. In a refined approach, pure PLA should be replaced by a PLA/bioglass (BG) composite, due to its advantageous biological and mechanical properties [55,56,61]. In addition, the antimicrobial potential of bioglass proven in vitro could be beneficial. The effectiveness of an biologically-adapted, bioprinted, and physiologically-enhanced scaffold concept for support of bone defect healing can now be further developed and verified in experimental studies.

## Figures and Tables

**Figure 1 materials-13-01836-f001:**
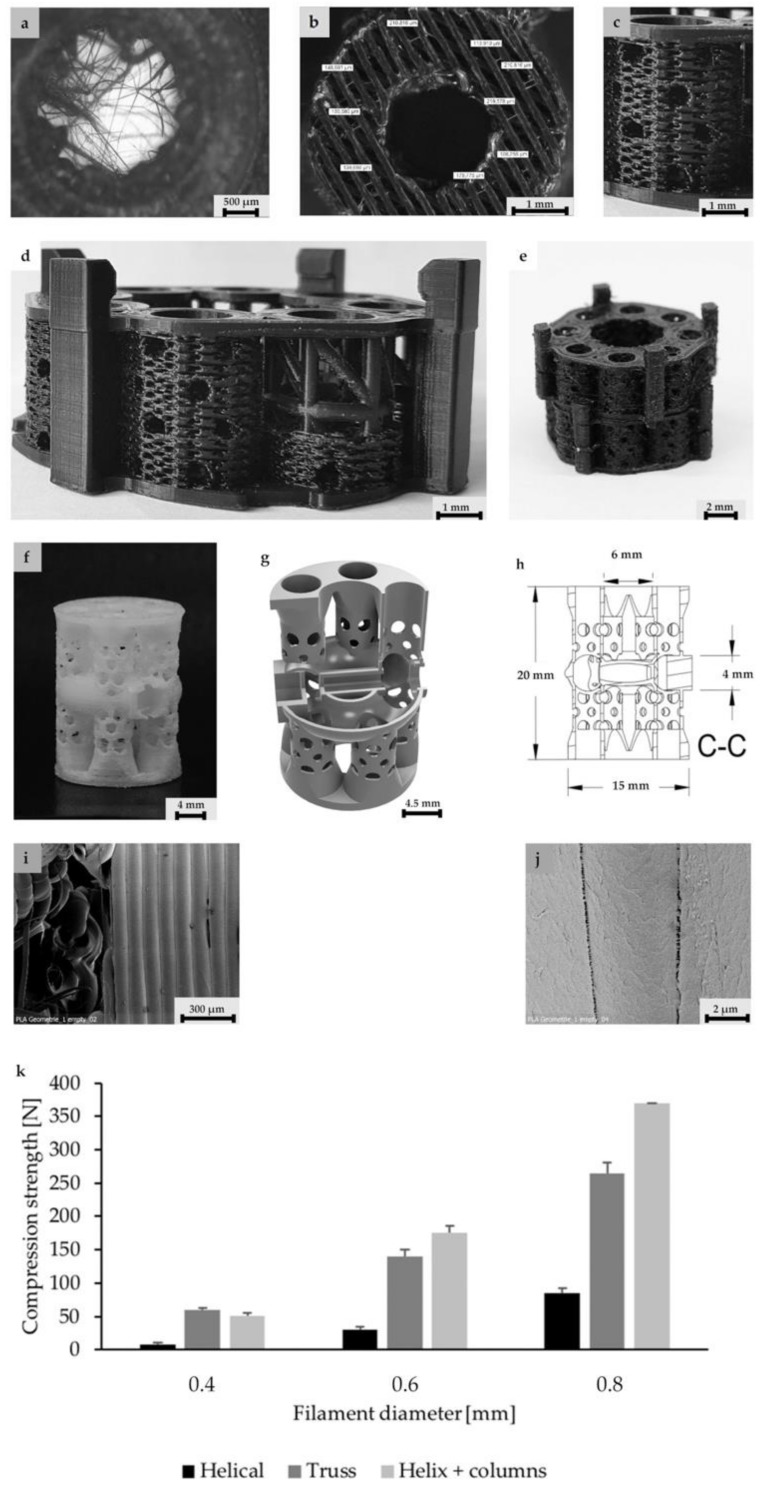
Structure levels in detail (for better illustration of structure details dyed PLA was used here): (**a**) Close up of a central column. In the center, the filamentary net is visible. Net printing via controlled extrusion, as used for surrounding wall printing, was not possible. Printing parameters had to be adjusted. Flow was decreased and distance between single filaments was increased from 0.1 to 0.3 mm. Althhough filamentary structures are discontinuous, they offer pores in the range of less than 100–150 µm and a high surface/volume ratio. (**b**) Axial image of single hollow column structure. For better illustration of the wall’s porous fine structure, the solid base ring is missing. (**c**) Basic unit of this concept: The single column image that inhabits the filamentary nets. The nets are easily accessible through the porous walls varying in diameter. (**c**,**d**) Eight hollow columns build up one subunit of the scaffold. At the bottom and top, base rings provide stability for a smooth force transduction. Additionally, the ring is the origin for filamentary Level 2 structures, shown in (**a**). (**c**,**d**) Subunits can be combined via an outsourced clamp system. The rigid truss-like structures were covered with porous walls, to ensure porosity in the range of 300–1500 µm. (**f**) 3D-printed design alternative: Design 2. (**g**) A highly porous structure with a central vertical and horizontal channel as well as four rotationally symmetric hollow cylinders. The structure consisted of columns functioning as subunits, which were combined by means of a small pedestal with a ring on the bottom and the top. (**h**) Dimensions of the second design. (**i**,**j**) Microstructures of untreated PLA scaffolds in SEM. Intercolumn areas of the filaments offer narrow spaces (**i**) and tight cooling cracks (**j**). (**k**) Verification of the axial compressive strength of three structural designs and characteristic diameter of struts vs. compressive strength. Test were performed with single hollow columns (diameter: 5 mm; height: 5 mm) (**c**). Finally, a compromise between strength and material volume was chosen. Experiments were carried out only once for verification of the design. In the future, a validation of the structure with the final design must be carried out. To adapt a scaffold to the size of a critical size bone defect, any number of hollow columns can subsequently be combined on a common base plate. The load is then distributed evenly.

**Figure 2 materials-13-01836-f002:**
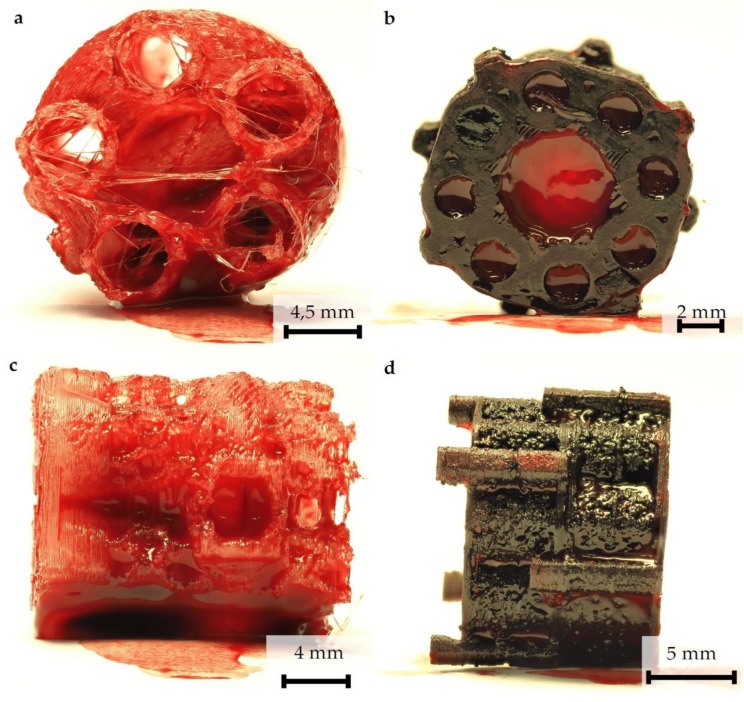
Blood penetration and hematoma formation in both scaffold designs: Each scaffold was immersed in untreated native blood for 30 s. (**a**) Axial view of design 2 after removal of the frontal cover. (**b**) Axial view of design 1. (**c**) Side view of design 2. (**d**) Side view of design 1. Two scaffolds were combined. The connection via an outer clamp system is clearly visible. The photos were taken after 3-h incubation. A complete wetting of all Scaffold surfaces can be seen. Hematoma formation in both scaffolds is visible in the large vertical (**a**,**b**) and horizontal (**c**,**d**) channels.

**Figure 3 materials-13-01836-f003:**
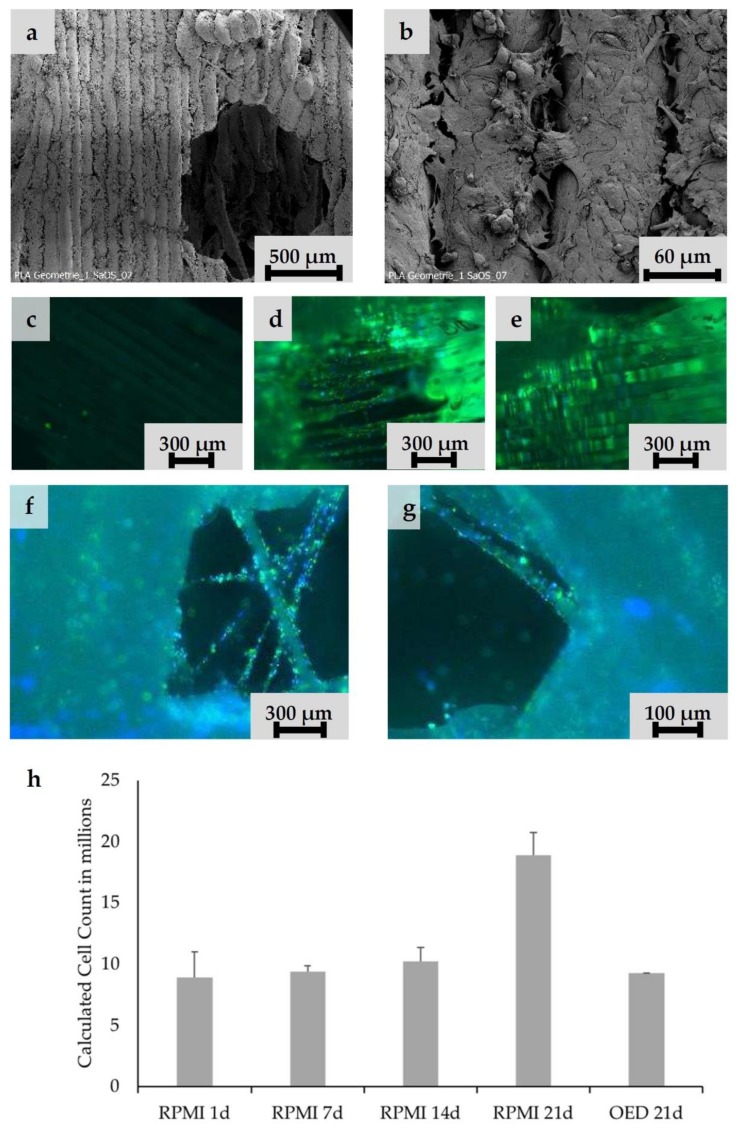
Cell adhesion experiments. Saos-2 cells seeded on undyed PLA scaffolds pretreated with adhesion protein coating: (**a**,**b**) Ingrowth of Saos-2 cells into the layers intercolumn and macroscopic pores is visible. (**c**,**d**) CFSE staining of native and *CellTak* pretreated PLA surfaces. (**c**) Native surface without preconditioning. Only a few cells were adherent. (**d**,**e**) *CellTak*-preconditioned surfaces showed increased cell adhesion. (**f**,**g**) CFSE/DAPI staining of Saos-2 cell coated scaffold (Design 2); cells appear green (CFSE) with blue nucleus (DAPI). (**f**) Stringing in a pore covered with vital cells. (**g**) Onsight on a pore with stringing structure. On all structural features, multiple vital cells are visible. (**h**) Calculated cell counts based on MTT-Test results after 1, 7, 14, and 21 days.

**Figure 4 materials-13-01836-f004:**
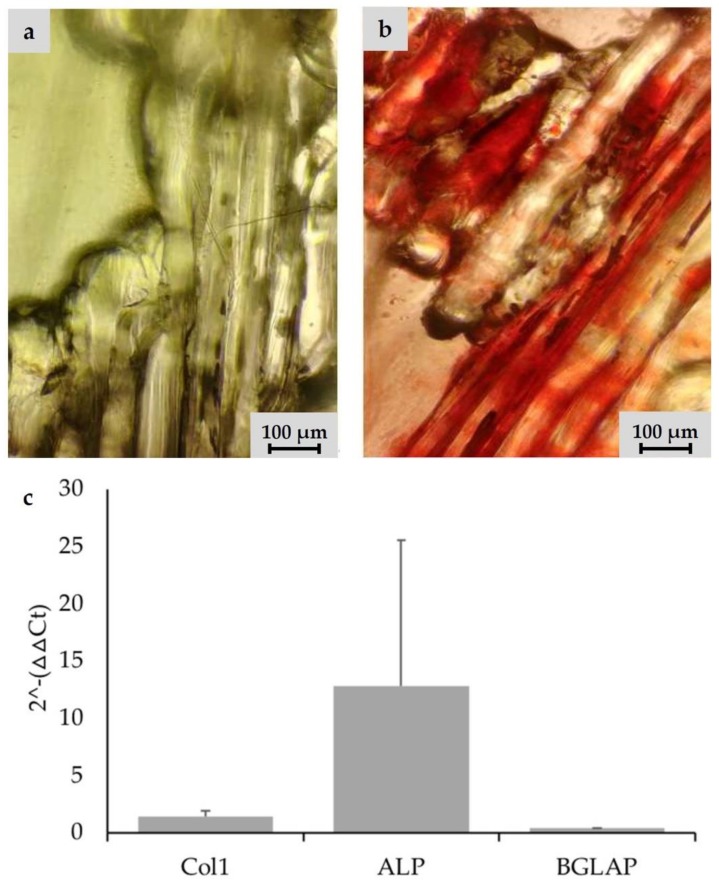
Detection of calcium deposition via alizarin staining on undyed scaffold prototype coated with Saos-2 cells after 21 days: (**a**) left scaffold was seeded in RPMI-medium; (**b**) right scaffold was incubated with osteogenic differentiation medium; and (**c**) PCR results for osteogenic differentiation on Day 21. Saos-2 cells seeded on scaffolds were incubated with RPMI medium and osteogenic differentiation medium. Focus was on cDNA for osteogenic proteins: collagen 1 (Col1), alkalic phosphatase (ALP), and osteocalcin (BGLAP). GAPDH was used as internal standard. The 2^−(ΔΔCt)^ values are presented.

**Table 1 materials-13-01836-t001:** Design concept of hierarchically organized levels for modular scaffold assemblies (explanation in text). For better illustration of structure details, dyed PLA was used here.

Level	Description	Pore Size	Visualization	Properties
1	PLA and internal porosity	1–10 µm		Osteoconductive Resorbable Available disinfectable
2	Microfilamentary mesh	≤150 µm	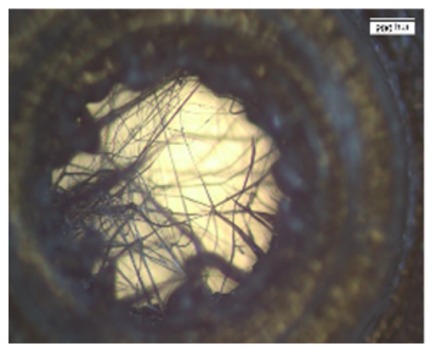	Osteoinductive Early resorption
3	Frame with interconnected pores	0.15–5 mm	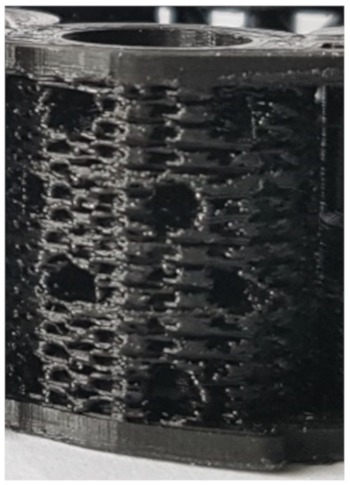	Supplementation Access Nutrition supply exchange Gas exchange
4	Tube assembly with central cavity	15–20 mm	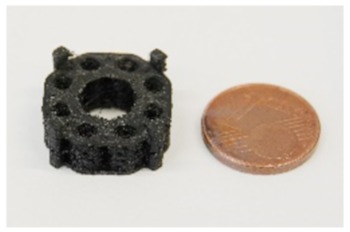	Mechanical stability Hematoma formation
5	Clamp for connectivity	–	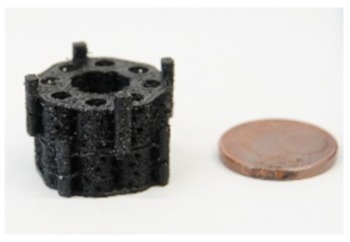	Modular extendable Customizable

**Table 2 materials-13-01836-t002:** Optimized printing parameters for filamentary net structures compared to wall printing parameters.

	Temperature (°C)	Speed (mm/s)	Flow (%)	Infill Line Distance (mm)	Infill Pattern
Walls	195	10	160	0.1	zig zag
Mesh	195	10	30	0.3	zig zag

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
