# Peer review of "3D-Printing of Hierarchically Designed and Osteoconductive Bone Tissue Engineering Scaffolds"

_materials, 2020, doi:10.3390/ma13081836_

Round 1

Reviewer 1 Report

This article is focused on introducing 3D-printing of hierarchically designed and osteoconductive bone tissue engineering scaffolds using Polylacetic acid (PLA). Article points out important points such as ‘a greater attention must be paid to the geometric design of scaffolds and focuses on the high porosity and mechanical strength of the introduced scaffold. However, this article introduces ideas that are not likelihood for tissue engineering and lacking in evidence. 

Concerns:

  1. Line 342-343 - The article mentions that using PLA shows good healing results in small animals but transfer to large animals is difficult (line 17-18, 31). To mention difficulty in ‘upscaling’ in the article, there should be more proof that 3D printed fused filament fabrication brought out better results in larger animals instead of stating ‘the scaffold could easily be scaled up and transferred to human size…’
  2. There are many statements in this article that lacks evidence to support the claims.
  3. Line 345-351 - In the last paragraph of Process Development, the article states that all individual systems were evaluated for robustness, rotational stability, flexibility, flexural strength, torsional stability, printability, effect on the bioactive structures and reversibly of the locking mechanism. Need to provide data to prove that all those tested criteria showed promising results.
  4. Line 421-422 – no evidence is provided to prove that the scaffold’s design offers required axial pore orientation for cell invasion that would furthermore perpendicular pores, resulting in facilitation in angiogenesis. 
  5. Line 417-418 – more explanation and data support are required to support that stimulation of angiogenesis and osteogenesis was incorporated into the design concepts in this paper. 
  6. Line 358 – need to provide evidence that hematoma formation will successfully form within the scaffold
  7. Line 377 – The claim that scaffold degradation will be replaced by natural bone has no support. There has to be a proper or controllable degradation rate associated with bone regeneration cascade. 
  8. Line 286 – Smooth PLA preconditioned surface is something that cannot easily be done in clinical application. Also, certain ideas such as adding osteogenic substances into the culture medium to induce osteogenic differentiation to Saos-2 cells is not something innovative but well-known with in the field.
  9. Can the scaffold actually carry out the cells fully throughout?
  10. Figure 1a – The necessity for microfilaments are understandable since the pore size is too large to be filled but can cells really attach to these tiny filaments (Line 232-233)? Also, can the filament formation be controlled? From the image provided, it does not seem to be a controlled environment. 
  11. Figure 2a – the pore, about 1mm in diameter, does not have any cells attached. How can this gap be filled in?

Reviewer 2 Report

I really enjoyed reading your paper. A great deal of work clearly went into this. Thank you for sharing it. I just have 5 comments:

  1. The paper would benefit from more specific information about the filament material used. From the images in Table 1 it appears that you used black colored PLA. The black color dye has an unpredictable negative effect on cells. Please consider using undyed PLA.
  2. The paper would benefit from a perfusion study or at least a software simulation to characterize the nutrition diffusion through the scaffold to better support the design.
  3. More information about the dimensions of the scaffold used in the mechanical test and an analysis of the modulus of the scaffold would be extremely helpful in understanding the benefits of this design.
  4. Is Figure 1 b a true representative image of the the scaffold matrix? There seems to be poor deposition from the printer.  Is it a reproducible design?
  5. The paper would benefit greatly from some statistical analysis as there is none presented.

Please don't let these critiques dissuade you from continuing with your excellent work.

Round 2

Reviewer 1 Report

All comments are well addressed.